# Quantification of Female Chimeric Cells in the Tonsils of Male Children and Their Determinants

**DOI:** 10.3390/cells12162116

**Published:** 2023-08-21

**Authors:** Boris Dmitrenko, Vincent Gatinois, Morgana D’Ottavi, Ahmed El Mouatani, Pauline Bouret, Ségolène Debiesse, Michel Mondain, Mohamed Akkari, Maxime Dallemagne, Franck Pellestor, Philippe Vande Perre, Jean-Pierre Molès

**Affiliations:** 1Pathogenesis and Control of Chronic and Emerging Infections, University of Montpellier, INSERM, EFS, 34394 Montpellier, France; borisdmitrenko@orange.fr (B.D.); morgana.d-ottavi@inserm.fr (M.D.); segolene.debiesse@inserm.fr (S.D.); p-van_de_perre@chu-montpellier.fr (P.V.P.); 2Unit of Chromosomal Genetics and Research Plateform Chromostem, Department of Medical Genetics, Arnaud de Villeneuve Hospital, Montpellier CHRU, 34090 Montpellier, France; v-gatinois@chu-montpellier.fr (V.G.); ahmed.elmouatani@outlook.com (A.E.M.); p-bouret@chu-montpellier.fr (P.B.); f-pellestor@chu-montpellier.fr (F.P.); 3Department of ENT and Head and Neck Surgery, Gui de Chauliac Hospital, University of Montpellier, 34295 Montpellier, France; m-mondain@chu-montpellier.fr (M.M.); m-akkari@chu-montpellier.fr (M.A.); m-dallemagne@chu-montpellier.fr (M.D.)

**Keywords:** microchimerism, cell trafficking, FISH, tonsil, breastfeeding, automated microscopy scanning

## Abstract

The factors influencing mother-to-child cell trafficking and persistence over children’s lives have yet to be established. The quantification of maternal microchimerism was previously reported through HLA-based approaches, which introduced bias regarding the tolerogenic environment. We aimed to identify cells of maternal origin irrespective of the HLA repertoire and to ascertain the determinants of microchimeric cells. This case–control study enrolled 40 male infants attending pediatric surgery from January 2022 to October 2022. Female cells were quantified in infants’ tonsil tissue by using cytogenetic fluorescent in situ hybridization (FISH) coupled with optimized automated microscopy. Out of the 40 infants, half (47.4%) had been breastfed for more than one month, a quarter for less a month, and 10 children (26.3%) were never breastfed. XX cells were observed in male tonsils in two-thirds of participants at a median density of 5 cells per 100,000 cells. In univariate analyses, child age was negatively associated with a high female cell density. In exploratory multivariate analyses, previous breastfeeding is a likely determinant of the persistence of these cells in the host, as well as the rank among siblings. Part of the benefit of breastmilk for child health may therefore be driven by breastfeeding-related microchimerism.

## 1. Introduction

The mechanisms underlying the benefits of breastmilk for child health have yet to be completely elucidated. The NeoVita Study clearly demonstrated, in a large meta-analysis, that breastfeeding initiated as early as the first hours of life is linked to a substantial reduction in child mortality [1]. Furthermore, breastfeeding reduces child morbidity, the risk of childhood infection [2], the risk of cancer [3], and nutrition-related harms to cognitive development [4]. Breastmilk is the most appropriate diet for a baby in terms of nutriments, antibodies, and other soluble elements [5,6]. In addition, other physiological processes may be implicated, such as the transfer of maternal cells to the offspring and maternal cell retention throughout infant life. Indeed, breastmilk contains maternal cells primarily consisting of epithelial cells, but also of immune cells and of stem cells at a ratio of up to 6% [7].

During pregnancy, bidirectional cell trafficking occurs between the mother and child. The processes of these foreign cells becoming resident in the new hosts are termed maternal and fetal microchimerism. Many investigations have estimated the density of microchimeric cells to host cells at about 1:3000–5000. The contribution of breastfeeding to microchimerism has more recently been demonstrated in animal models either as an additional route of mother-to-child cell trafficking [8] or as promoting long-term residency of pregnancy-related microchimeric cells by increasing the tolerogenicity of the host immune system [9,10]. Although microchimeric cells likely participate in organ maturation (immune, brain) and cell regeneration, they may also have implications in cancer or autoimmune diseases [11,12,13].

Cell traffic is highly likely in humans considering the cellular composition of colostrum, the reduced salivary protease activity, the more neutral pH of gastric secretions in newborns and infants, as well as the gut permeability during the first few weeks of life or in the case of gut inflammation [14]. In mouse models, the tolerance to immunologically foreign non-inherited maternal antigens (NIMA) is driven by a systemic accumulation of maternal immunosuppressive regulatory T-cells (Tregs) with NIMA specificity [11,15]. More recent findings suggest that postnatal depletion of maternal cells modified the maturation of natural killer cells and T-lymphocytes [16]. In humans, infants with detectable maternal microchimerism at birth were shown to have an improved polyfunctional CD4+ T-cell response to the BCG vaccine [17]. Recently, Harrington’s group described exclusive breastfeeding as a determinant of higher levels of microchimeric cell density (adjusted RR 4.05; 95% CI, 0.85–19.44, *p* = 0.080) [17], but has not yet been able to discriminate cell trafficking from tolerogenic activity. Precise measurements of microchimeric cell density per organ and its evolution over time are lacking in both animal and human models.

Most animal models developed to study microchimerism involved cross-fostering experiments [8,18,19,20], a situation which is very infrequent in humans to demonstrate these processes. In humans, studies use molecular approaches for maternal cell discrimination from host cells and most, if not all, target the HLA repertoire. Mother–child dyads with a compatible HLA repertoire may have different densities and rates of retention of maternal microchimerism.

In the present study, we identified XX cells of maternal origin in tonsils from young males by using fluorescent in situ hybridization (FISH) identification of sex chromosomes in male samples, coupled with an optimized automated microscopic quantification. Breastfeeding-related microchimeric cells could be detected in any organ including the brain [20], but the tonsil was selected for this study given its accessibility from the pediatric theatre. Our objectives were (i) to perform a quantitative analysis of the microchimeric cell counts and (ii) to investigate the possible association with feeding modalities in early infancy.

## 2. Materials and Methods

### 2.1. Study Design and Participants

This was an observational pilot study performed from January 2022 to October 2022. The study participants included 40 young males attending the pediatric surgery department of the Montpellier Teaching Hospital. The inclusion criteria included: (i) being male, (ii) being aged between 1 and 16 years, and (iii) being scheduled for a tonsillotomy or tonsillectomy for medical reasons in accordance with national guidelines [21]. The non-inclusion criteria included: (i) an unknown breastfeeding status during infancy, (ii) preterm birth (gestational age under 34 weeks), (iii) a history of organ or bone marrow transplant, (iv) the known existence of a twin (male or female), and (v) the known existence of a gonosomal anomaly. Two female patients were also included to establish the performance of the technique.

### 2.2. Sample Collection and Processing

Prior to surgery, a medical investigator met the parents or guardians who provided informed consent for their child’s participation. A short questionnaire related to socio-demographic variables, delivery routes, and breastfeeding practices was administered to the parent/guardian. Tonsil samples were obtained from the pediatric department, transferred to the laboratory, and handled within 6 h. Each sample (left and right) was cross-sectioned along its longest axis. Half was stored at −20 °C, and the second half was smeared on glass slides (*n* = 10) and stored at −20 °C until further use. Data and samples were collected using a unique study identification number, maintaining confidentiality throughout.

### 2.3. Detection of X and Y Chromosomes by FISH

Tonsil smears were processed in accordance with a previously published FISH protocol, which included pre-treatment, protease treatment, hybridization, washing, and counterstaining steps [22]. We used a mix of probes, consisting of the specific centromeric probe of chromosome X (DXZ1 labeled with FITC), the specific centromeric probe of chromosome Y (DYZ3 labeled with Texas Red), and the specific centromeric probe of chromosome 18 (D18Z1 labeled with AquaSpectra) (Aquarius Fast FISH Prenatal kit, Cytocell, Cambridge, UK). FISH hybridization was performed in accordance with the manufacturer’s recommendations.

### 2.4. Automated FISH Analysis

All slides were scanned using the automated slide scanning platform METAFER/Metacyte© (Metacyte, Altlußheim, Germany) coupled with a fluorescent microscope (Zeiss, Axio Imager.Z2, Iena, Germany). A multistep reading algorithm was designed in order to analyze a minimum of 8000 nuclei per patient, merging nuclei from both the left and right tonsils. Firstly, interpretable interphase nuclei were identified by deleting clusters, artefacts, and stain debris. Second, based on DAPI staining, contiguous nuclei were automatically separated. Each identified nucleus was stacked and z-scanned (*n* = 3 focal planes) at the three color wavelengths. Nuclei were then automatically classified according to the number of red, green and blue spots. After this primary automated acquisition, all female nuclei candidates (two XX signals) underwent reclassification and manual curation by two experienced operators, blind to each other’s results. Nuclei with conflicting classifications were ultimately reassessed by an experienced third operator. Finally, we used the ratio of the total number of detected female nuclei to the total number of scanned nuclei for each participant’s left and right tonsils combined. A full set of quality controls are presented in the Appendix B.

### 2.5. Statistical Analyses

Categorical variables were presented as “count (%)”, while continuous variables including socio-demographic or cell count results were expressed using “median [IQR]” or “mean (min–max)”, respectively. A comparative logistic regression model was used to evaluate the microscopic quantification technique. Inter-observer variability was evaluated using Spearman’s correlation coefficient. Categorical and continuous socio-demographic characteristics were compared using Fisher’s exact test and Wilcoxon’s rank sum-test, respectively.

Factors associated with female cell detection (dichotomous outcome variable) and female cell density among patients for whom female cells were detected (continuous outcome variable) were assessed, first in a univariate analysis using Fischer’s exact test and univariate logistic regression for the dichotomous outcome variable. The results are presented as counts and row percentages or odds ratios with their 95% CI. For the continuous outcome variable, Spearman’s correlation coefficient was used for continuous parameters, Wilcoxon’s rank sum test was used for categorical parameters in two categories, and the Kruskal–Wallis equality of populations rank test was used for categorical parameters in more than two categories. The results are presented as row medians or correlation coefficients.

Finally, in order to explore possible interactions or combined effects of the parameters, the continuous outcome variable (female cell density) was analyzed using a multivariate Poisson regression model with a robust standard error correction to account for the small sample size.

All statistical analyses were performed with GraphPad-Prism version 8.0.2 (GraphPad Software, Boston, MA, USA), Stata 16.1 (Stata Corp, College Station, TX, USA), and RStudio 4.2.2 (Posit Software, PBC, Boston, USA). The threshold for statistical significance was set at *p* < 0.05.

## 3. Results

### 3.1. Patient Recruitment and Characteristics

Two of the forty participants were excluded from analysis. One was erroneously included as he was identified as having a twin sibling after inclusion. The second exclusion was due to the automated cell readings never reaching the established minimum nuclei number set at 8000 nuclei per patient.

The characteristics of the 38 analyzed males are shown in Table 1. The median age for included children was 4 years [IQR: 3–6], and two-thirds were considered to have a normal BMI for their age. Roughly half (47.4%) had been breastfed for more than one month, and 10 children (26.3%) were never breastfed. The demographic and clinical characteristics of the breastfed versus never-breastfed children did not differ significantly (Appendix A).

### 3.2. Female Cell Density in Young Boys’ Tonsils and Its Associated Factors

We identified a total of 48 XX nuclei (Figure 1) for a total of 1,166,308 scanned nuclei. On average, 30,692 nuclei were scanned per patient (min–max: 8665–70,313). XX nuclei were identified in 57.9% (22/38) of patients. Amongst patients with identified XX nuclei, we found an average of 2.18 XX nuclei/patient (min–max: 1–13) for a mean of 33,936 nuclei screened per patient (min–max: 10,018–70,313). Altogether, XX nucleus density ranged from 2.3 to 21.2 per 100,000 analyzed nuclei, with a median XX nucleus density of 5.0 per 100,000 [IQR: 3.6–7.2].

Characteristics associated with the detection of female nuclei are described in Table 2. The child’s age at the time of the surgery appears to be moderately associated with female nucleus detection, but this is likely a bias from the nature of the convenience sampling with highly variable ages and female nucleus detection in the only two teenage participants in our sample. Neither having been breastfed nor feeding modality were associated with female nucleus detection.

Among patients in whom female nuclei were detected, the child’s age at the time of the surgery was significantly and inversely correlated with female nucleus density, with 0.5 fewer microchimeric cells per 100,000 cells read for each one-year increase in age (Table 3). The median female cell density per 100,000 cells read was 1.1 cells higher (22%) among children who received any breastmilk during infancy compared to those who were never breastfed (Table 3).

For those with no female cells detected in their sample, children were on average younger and breastfed for a shorter duration than those with identified female cells (Table 4).

Finally, multivariate analysis for factors associated with female cell density showed that, when accounting for child age, BMI, and the feeding mode during infancy, children who were not the first-born sibling had almost 2-fold the number of female cells per 100,000 cells read compared to children who were first-borns (*p* = 0.018) (Table 5). A significant interaction between child age and having been breastfed was observed. Female cell density significantly decreased with age on the whole, but the rate of female cell depletion was much faster for children that had never been breastfed, independently of BMI, sibling rank, or the duration of breastfeeding (Table 5, Figure 2). As a sensitivity analysis, removing the two oldest children from the analysis did not alter any of the model outputs (Appendix A).

## 4. Discussion

While the route of cell trafficking involved in fetal microchimerism is evidently transplacental [8], the routes involved in maternal microchimerism have yet to be fully established. Herein, we developed a robust experimental approach to identify rare female cells in young boys, regardless of the HLA repertoire. These cells were observed in male tonsils at a median density of 5 cells per 100,000 cells. In exploratory analyses, the long-term persistence of microchimeric cells appeared to be associated with breastfeeding which introduced another level of complexity to discriminate between cell trafficking occurring transplacentally or via breastfeeding, if any. Furthermore, the rank among siblings was also found to be a determinant of the density of microchimeric cells.

This FISH approach would not have been possible without automated scanning and analysis. Indeed, an appropriate compromise must be found between the number of analyzed nuclei and the number of z-scans per nuclei for optimized sensitivity and specificity of detection. Overall, automated acquisition time exceeded 24 h in almost half of the cases (mean analysis time was 20.4 h) with variable acquisition speeds, ranging from 4.04 to 57.2 nuclei/minute. The overall performance of this experimental approach has proven to be robust and reproducible, although optimizable. This combined technique could be adapted to the detection of rare events in other domains such as the detection of residual or metastatic cancer cells [23].

Female cells were identified in almost two-thirds of patients. The reported prevalence of microchimerism is consistent with previous reports ranging from 20% to 60% [17,24,25,26,27,28,29,30]. The reasons for the inability to detect female cells in some children may rely on the sensitivity of the technique. In the present study, the total number of cells to be analyzed was set a priori based on a review of published fetal microchimerism [8] and tonsil-specific data [24]. However, the pathological nature of these samples may have had an excess of inflammatory cells recruited on site which diluted the resident female cells. Alternatively, the tolerogenic activity of the host immune system is inoperative in a situation of a highly discordant HLA repertoire. Samples with no female cells detected might require extended investigations.

The female cell density among those with detectable female cells ranged from 2.3 to 21.2 female cells per 100,000. These numbers contrast with those found in the literature for fetal microchimerism [8], but were consistent with those reported recently for maternal microchimerism. The latter were obtained using HLA-specific PCR on blood DNA extract and averages varied from 0 to 153 genomic equivalents (gEq)/100,000 gEq [17,24,25,26,27,28,29,30], with individual maximum values up to 1200 gEq/100,000 gEq in a 15-week-old child [18] or 354 gEq/100,000 gEq in 22-year-old women [30]. With a cell density of about 1 maternal cell per 5000–10,000 cells, a complete description of maternal microchimerism for each organ and its dynamics over time is required to further improve knowledge in this field.

The routes of mother-to-child cell trafficking, known at least in animal models, include transplacental cell transfer during pregnancy, and the transfer of breastmilk cells across the infant’s gut surfaces [8]. We were not able to find for humans, such as in Balle’s study [17], a clear association between the female cell density and breastfeeding, even though microchimeric cell density was on average 22% higher in previously breastfed children compared to never-breastfed children. Furthermore, children who were not the first-born sibling were more prone to high female cell density, suggesting a more favorable fate of cells transmitted from the mother [31], but not reaching statistical significance as previously reported [32,33]. This high quantity of cells could result in a higher transmission rate, cell expansion or prolonged persistence due to a tolerogenic cell microenvironment. It was previously suggested that the first pregnancy might promote maternal tolerance to genetically similar future siblings [34]. In consequence, mother-to-child cell trafficking could be favored during ulterior pregnancies or breastfeeding periods. Another hypothesis that was not tested herein because the data were missing related to whether these cells originated from a previous female sibling through the “canonical route”. To our knowledge, no previous report mentioned cell transfer from sister to brother, but the reverse was observed in a Danish girl cohort, leaving this hypothesis open [35]. As previously reported by Balle et al., the level of female cell density varies throughout childhood [17]. Our exploratory analyses also found that being previously breastfed was associated with a gentle attrition of microchimeric cells with age, as compared to those that had never been breastfed. Given the relationship between breastfeeding and immune maturation [36,37], this observation suggests tolerogenic properties of microchimeric cells, such as for regulatory T-cells transferred from breastmilk [11], possibly resulting in enhanced future immune response and improved child health and development.

This study has several limitations. First, the number of samples obtained was relatively low, which limited the statistical power of the analyses. Multivariate analyses were therefore highly explorative and had the aim of supporting future investigations and helping the proper design of such studies. Second, the collected tonsil samples originated from patients with surgical indications. Consequently, these samples did not consist of normal tissues, and inflammatory processes likely altered their cellular composition. Third, the experimental approach can only be applied to male children, as the identification of maternal XX nuclei among daughters’ XX nuclei is impossible with our FISH assay. We suggest that conclusions drawn herein will be conservative in female children, since the mother–daughter immune conflicts are less pronounced. Finally, the infants’ feeding status was collected from interviews of the participants’ parents (or guardians), therefore potentially affected by recall bias.

## 5. Conclusions

An approach combining a routine FISH assay with an automated high-throughput scanning analysis allowed the identification of cell events as rare as 1 event in 5000–10,000, a sensitivity comparable to that of molecular approaches. This approach allowed us to identify and to quantify maternal microchimerism in secondary lymphoid organs from male children. The role of breastfeeding as an additional route for the microchimeric transfer of cells of maternal origin, and their persistence, deserves further investigation.

## Figures and Tables

**Figure 1 cells-12-02116-f001:**
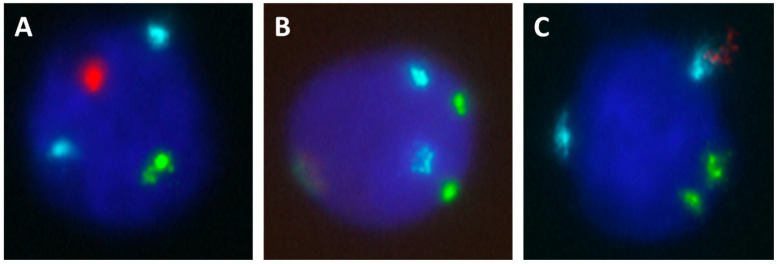
Examples of interphase nuclei after probe hybridization and classification. The same patient presented male cells (**A**), but also female cells (**B**,**C**). Red spot = Y chromosome, green spot = X chromosome and cyan spot = chromosome 18.

**Figure 2 cells-12-02116-f002:**
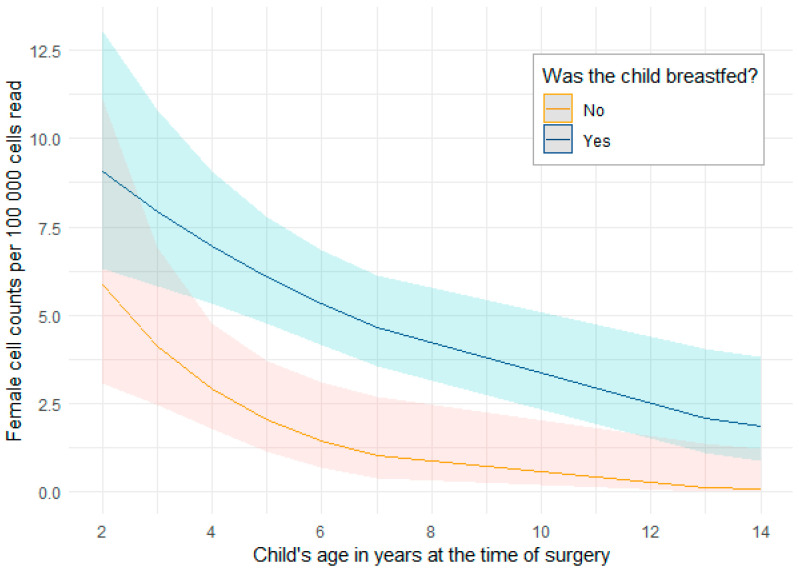
Fitted values and 95% CI from the final multivariate model for the interaction between having ever been breastfed during early infancy and child age, independently of BMI, sibling rank, and the duration of breastfeeding.

**Table 1 cells-12-02116-t001:** Characteristics of the analyzed participants (*n* = 38).

Characteristics	*n* (%) or Median [IQR]
Age of the child at the time of surgery, in years	4 [3–6]
BMI *, in kg/m^2^	15.8 [14.8–17.5]
Pre-term births (≤36 weeks)	
Yes	6 (15.8)
No	32 (84.2)
Breastfed	
Yes	28 (73.7)
No	10 (26.3)
Breastfeeding modalities ¤	
0	10 (26.3)
Colostrum (<5 days)	6 (15.8)
Transition milk (6 days to 1 month)	4 (10.5)
Mature milk (>1 month)	18 (47.4)
Breastfeeding duration, in days	21 [0–182.5]
First born	
Yes	20 (52.6)
No	18 (47.4)
Mother’s age at birth †	32 [27–33]

Abbreviations: IQR, interquartile range; BMI, body mass index; kg/m^2^, kilograms per meter squared. ¤ Children were categorized according to the duration of breastfeeding, * 2 missing values, † 1 missing value.

**Table 2 cells-12-02116-t002:** Participants’ characteristics associated with the detection of female cells.

Characteristics	Children with Detected Female Cells (*n* = 22)	
	*n* (%) or OR (95% CI)	*p*-Value
Age of the child, in years	1.5 (1.0; 2.2)	0.072
BMI *, in kg/m^2^	1.1 (0.9; 1.3)	0.375
Pre-term births (≤36 weeks)		0.370
Yes	5 (83.3)	
No	17 (53.1)	
Breastfed		0.713
Yes	17 (60.7)	
No	5 (50.0)	
Breastfeeding modalities ¤		0.444
0	5 (50.0)	
Colostrum (<5 days)	2 (33.3)	
Inter. milk (6 days to 1 month)	3 (75.0)	
Mature milk (>1 month)	12 (66.7)	
Breastfeeding duration, in days	1.0 (1.0; 1.0)	0.140
First born		0.512
Yes	13 (65.0)	
No	9 (50.0)	
Mother’s age at birth †	1.0 (0.8; 1.1)	0.691

Abbreviations: IQR, interquartile range; BMI, body mass index; kg/m^2^, kilograms per meter squared. ¤ Children were categorized according to the duration of breastfeeding. * 2 missing values, † 1 missing value.

**Table 3 cells-12-02116-t003:** Participants’ characteristics associated with the density of female cells.

Characteristics	Nb of Female Cells per 100,000 Cells Read (*n* = 22)			
	n	Median [IQR]	ρ	*p*-Value
Age of the child, in years	22		−0.502	0.017 °
BMI *, in kg/m^2^	22		−0.266	0.232
Pre-term births (≤36 weeks)				0.940
Yes	5	5.4 [2.8–11.8]		
No	17	4.7 [3.7–6.0]		
Breastfed				0.283
Yes	17	5.5 [3.7–7.2]		
No	5	4.4 [2.4–5.4]		
Breastfeeding modalities ¤				0.502
0	5	4.4 [2.4–5.4]		
Colostrum (<5 days)	2	3.8 [2.8–4.7]		
Inter. milk (6 days to 1 month)	3	5.6 [3.6–20.0]		
Mature milk (>1 month)	12	5.6 [3.9–9.5]		
Breastfeeding duration, in days	22		0.217	0.333
First born				0.324
Yes	13	4.4 [3.6–5.6]		
No	9	5.8 [3.7–11.8]		
Mother’s age at birth †	21		0.134	0.561

Abbreviations: IQR, interquartile range; BMI, body mass index; kg/m^2^, kilograms per meter squared. ¤ Children were categorized according to the duration of breastfeeding. ° Statistically significant, * 2 missing values, † 1 missing value.

**Table 4 cells-12-02116-t004:** Comparative characteristics of children according to female cell detection in the sample.

Characteristics	No Female Cells Detected	Female Cells Detected	
	*n* (%) or Median [IQR]	*n* (%) or Median [IQR]	*p*-Value
Age of the child, in years	3.5 [2–4.5]	4.5 [4–6]	0.038 °
BMI *, in kg/m^2^	16.2 [15.1–16.7]	15.3 [14.8–19]	0.808
Pre-term births (≤36 weeks)			0.370
Yes	1 (6.3)	5 (22.7)	
No	15 (93.8)	17 (77.3)	
Breastfeeding			0.713
Yes	11 (68.8)	17 (77.3)	
No	5 (31.3)	5 (22.7)	
Breastfeeding class			0.444
0	5 (31.3)	5 (22.7)	
Colostrum (<5 days)	4 (25.0)	2 (9.1)	
Inter. milk (6 days to 1 month)	1 (6.3)	3 (13.6)	
Mature milk (>1 month)	6 (37.5)	12 (54.6)	
Breastfeeding duration, in days	1.5 [0–65.6]	83.6 [0.5–182.5]	0.107
First born			0.512
Yes	7 (43.8)	13 (59.1)	
No	9 (56.3)	9 (40.9)	
Mother’s age at birth †	32.5 [27.5–33.5]	31 [27–33]	0.517

Abbreviations: IQR, interquartile range; BMI, body mass index; ° Statistically significant, * 2 missing values, † 1 missing value.

**Table 5 cells-12-02116-t005:** Final multivariate model for female cell density, per 100,000 male cells (*n* = 22).

Characteristics	IRR (95% CI)	*p*-Value
Child age, in years	0.71 (0.60; 0.82)	<0.001 °
Breastfeeding		0.988
Yes	1.01 (0.47; 2.17)	
No	Ref.	
Interaction breastfeeding * child age	1.24 (1.05; 1.47)	0.012 °
First born		0.018 °
Yes	Ref.	
No	1.82 (1.11; 2.98)	
Child BMI, in kg/m^2^	1.03 (0.99; 1.08)	0.101
Breastfeeding duration, in days	1.00 (1.00; 1.00)	0.166

Based on a Poisson regression, with a breastfeeding and child age interaction term and adjusted for sibling rank, BMI, and breastfeeding duration. Abbreviations: IRR, incidence rate ratio; CI, confidence interval; BMI, body mass index; kg/m^2^, kilograms per meter squared. ° Statistically significant.

## Data Availability

The data presented in this study are available on request from the corresponding author.

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
