# Peer review of "Quantification of Female Chimeric Cells in the Tonsils of Male Children and Their Determinants"

_cells, 2023, doi:10.3390/cells12162116_

Round 1

Reviewer 1 Report

This study describes a method to quantify maternal cells present within male offspring from tonsil samples obtained during surgery.  The method used is FISH and has only been possible with the improvements made in automated microscopy for screening thousands of nuclei.

I read this study with much interest and find it to be eloquently described with all of the expected limitations well laid out.  Despite the participant number limitations the authors should be commended for conducting suich a difficult study.  It is very difficult to recruit just the right patient types to study microchimerism.  It is an extremely reclusive phenomenon and as the authors point out could be easily missed if the wrong patient age was chosen.

My only wish for the study was that the samples that were positive by FISH had also been tested using traditional qPCR to enable those of us who have only described microchimerism using qPCR to have confidence that those methods are detecting microchimeric cells to the same rate that visual methods do.  Alas i do understand how challenging it would be to obtain the appropriate matched samples to conduct this aspect.

Author Response

Thank you for your comments.

The protocol submitted to the Ethics Committee for approval did not include taking a maternal blood sample. Therefore, it was a real challenge to compare the techniques because we could not select a marker to identify the X chromosome that was not transmitted to the child. But we asked ourselves this question and tried to estimate the proportion of inactivated X chromosomes using molecular biology, by testing methylation products. These attempts were not successful. As the reviewer points out, it was difficult for us to amend the protocol a year after recruitment ended to include mothers sampling. A lesson for the next study.

Reviewer 2 Report

Dmitrenko et al. surveyed a cohort of 40 male children regarding the detection of female cells in resected tonsils using FISH analysis. Results were accompanied by data on the children having been breastfed during infancy, obtained by questionnaires. Despite the small size of the cohort, breastfeeding showed to be a positive influence of the occurrence of female cells in tonsil tissue.

Overall, the manuscript is well written, mostly organized, and comprehensible. Nevertheless, there are some minor concerns with this report:

1.       Table titles and legends

a.       Tables 1+2+3 have the same title, which is misleading for table 3, in which only the characteristics of mc+ patients are displayed and not of all patients (n=22, not 38).

b.       Table 3 completely misses a legend/abbreviations

c.       Table 4 misses the description for *

2.       Female cell density

a.       Of the 22 participants with detectable female cells 59% were first born (supplementary table 2). Yet, in not-first-born patients almost double the amount of female cells could be detected. How do the authors rule out the possibility of older female siblings’ cells? If there is data on the older siblings’ gender, the study could benefit from including an analysis of not-first-borns with and without older female siblings.

b.      Beside of the origin, the nature of the microchimeric cells would be of great interest. Are the microchimeric cells rather T cells (regulatory, naïve CD4 or CD8, …), B cells or other?

c.       The authors state that “Female cell density significantly decreased with age on the whole, but the rate of female cell depletion was much faster for children that had never been breastfed, independently of BMI, sibling rank, or the duration of the breastfeeding (Table 4, Figure 2)” (lines 208-211). The model for non-breastfed patients has been calculated from only 5 patients, and the timeline spans 12 years. Are these depletion models truly comparable?

3.       Manuscript title

a.       For the most part of the manuscript the authors precisely describe “female cells” and not “maternal cells”. As long as it cannot be ruled out that the origin of the female cells might be of older sisters, the title should be revised

4.       References

a.       References 17 and 18, as well as 7 and 14 are identical

b.       Less than one third of references are from the past 5 years (2018-today); please include more recent publications

Author Response

Thank you for your comments.

Table titles and legends

  1. Tables 1+2+3 have the same title, which is misleading for table 3, in which only the characteristics of mc+ patients are displayed and not of all patients (n=22, not 38).
  2. Table 3 completely misses a legend/abbreviations
  3. Table 4 misses the description for *

We apologize for not checking carefully the transfer to the template and thanks for pointing this out. This is now corrected. * is this table is not a reference to a footnote but shows the interaction between two variables. For the seek of clarity we added spaces between the symbol and the name of the variables. Other authors used x or # as symbol.

Female cell density

  1. Of the 22 participants with detectable female cells 59% were first born (supplementary table 2). Yet, in not-first-born patients almost double the amount of female cells could be detected. How do the authors rule out the possibility of older female siblings’ cells? If there is data on the older siblings’ gender, the study could benefit from including an analysis of not-first-borns with and without older female siblings.

This hypothesis is indeed not ruled out, however we do not have the data to test it. We added to the discussion section the following text: Line 311-315 (document with tracking) “Another hypothesis that was not tested herein because the data were missing, would origin these cells from previous female sibling through the “canonical route”. To our knowledge, no previous report mentioned cell transfer from sister to brother but the reverse was observed in a Danish girl cohort, leaving the hypothesis open [35]”

.

  1.     Beside of the origin, the nature of the microchimeric cells would be of great interest. Are the microchimeric cells rather T cells (regulatory, naïve CD4 or CD8, …), B cells or other?

FISH technique as performed in this study were not coupled with immunohistochemistry. We cannot provide an answer to this question.

  1. The authors state that “Female cell density significantly decreased with age on the whole, but the rate of female cell depletion was much faster for children that had never been breastfed, independently of BMI, sibling rank, or the duration of the breastfeeding (Table 4, Figure 2)” (lines 208-211). The model for non-breastfed patients has been calculated from only 5 patients, and the timeline spans 12 years. Are these depletion models truly comparable?

We understand the questioning and we shared it. First, the time-dependant model used cross-sectional data, which, we agree, is not the best design. However, it will be impossible to get longitudinal data for tonsil tissues, after their resection. The small sample size oriented us to use the statistical tools that were the most adapted i.e. Poisson regression model with Huber and White robust variance estimator. To challenge the model for the range of time, we also performed a sensibility analysis by removing the two oldest participants to prevent from imbalanced follow-up duration between the groups. These oldest participants belonged to the group of breastfed infants. The final outputs remained unchanged. All these limitations are mentioned in the text (see Line 239-241; 325-328).

Manuscript title

  1. For the most part of the manuscript the authors precisely describe “female cells” and not “maternal cells”. As long as it cannot be ruled out that the origin of the female cells might be of older sisters, the title should be revised

We agree with the reviewer and we changed the title accordingly. New title “Quantification of female chimeric cells in tonsils of male children and its determinants”

References

  1. References 17 and 18, as well as 7 and 14 are identical

Thank you for pointing this out. We corrected the reference list and the reference numbers

  1. Less than one third of references are from the past 5 years (2018-today); please include more recent publications

We updates few references but we should admit that the field is not “vigorous” in term of recent publications (about 30 references each year since 2018 with maternal microchimerism PubMED).

Reviewer 3 Report

The text is very interesting. The abstract, introduction and patients and methods part are concise.

The results section could be enriched with the inclusion of tables S1 and S2.

The discussion has finalized on the present paper results,

Problems

1) how can the authors be certain that the difference in the frequency of female cells in the context of male tonsil tissue is due to the effect of mother's milk? Table 2 shows that infants with maternal cell detection do not have a statistically significant difference compared to formula-fed infants,

2) The FISH technique is much less sensitive than RT-PCR, the authors should explain why they preferred FISH

3) how can the authors explain the negativity of maternal cells in younger children as shown in table S2?

4) table 1 shows "at term" children with 37 weeks, while in the following tables >36 weeks is reported, the authors could use the same definition,

The English language requires minimal corrections

Author Response

Thank you for your comments.

The results section could be enriched with the inclusion of tables S1 and S2.

We have added the supplementary table S2 to the text as these data help the understanding of the manuscript. We did not added the supplementary table S1 in order to keep an easy read of the manuscript, the information within this table is part of an exhaustive analysis and is of interest for advanced reader from the field.

The discussion has finalized on the present paper results,

Problems

  • how can the authors be certain that the difference in the frequency of female cells in the context of male tonsil tissue is due to the effect of mother's milk? Table 2 shows that infants with maternal cell detection do not have a statistically significant difference compared to formula-fed infants,

Some sentences were probably not worded correctly, making the message ambiguous. We removed some of them: Line “257-260 “The total number of microchimeric cells reported herein accounted for both transplacental and breastfeeding cell trafficking, the placental route being more effective and representing three quarters of these” and Line 342-343 “The proportion of cell transfer from mother to child during breastfeeding is most probably too low to be quantified accurately with this sampling” these sentences were removed.

However, we clearly stated the following :

Line 194-195: “Having been breastfed or modality of breastfeeding was not associated with female nuclei detection.”

Line 283-286: “We were not able to find, such as in Balle’s study [18], a significant association between the female cell density and breastfeeding although microchimeric cell density was on average 22% higher in previously breastfed children compared to never-breastfed children.”

Line 289-290: “This high quantity of cells could result in a higher transmission rate, cell expansion or prolonged persistence due to a tolerogenic cell microenvironment.”

If any other sentences need revisions please help us to identify them?

  • The FISH technique is much less sensitive than RT-PCR, the authors should explain why they preferred FISH

We agree with the reviewer that classical FISH technique is less sensitive than PCR. We believe that the pipeline presented herein has an improved sensitivity, nearing that of a PCR. Molecular biology approaches required the identification of markers of the non-transmitted X chromosome, whereas the FISH approach does not require such markers. Furthermore, most of the markers are selected from the HLA repertoire, which introduces a selection bias. Finally, FISH the technique may be applied to tissue sections, which, after validation, will provide additional information such as localization of MC cells within the tissue, type of the MC cells if coupled with immunohistochemistry.

  • how can the authors explain the negativity of maternal cells in younger children as shown in table S2?

Other reports showed the lack of microchimeric cells detection in some individuals. Some reasons were developed in the discussion section from Lines 263 to 270. Whether they are present or not is highly related to the threshold of the technique and the prevalence of said cells. We mentioned two situations that could lower the prevalence, one “artificially” with the dilution with the inflammatory cells, the second with a potential mother-child immune conflicts. We can also cite that grandmother microchimeric cells were not detected systematically in the cord blood of grandchild (Karlmark KR Grandmaternal cells in cord blood. EBioMedicine. 2021 Dec;74:103721. doi: 10.1016/j.ebiom.2021.103721. Epub 2021 Nov 26).

  • table 1 shows "at term" children with 37 weeks, while in the following tables >36 weeks is reported, the authors could use the same definition,

The text was modified accordingly.

Round 2

Reviewer 3 Report

The authors fully replied to all criticism

The english language is fine